# Reduction of Fat to Muscle Mass Ratio Is Associated with Improvement of Liver Stiffness in Diabetic Patients with Non-Alcoholic Fatty Liver Disease

**DOI:** 10.3390/jcm8122175

**Published:** 2019-12-09

**Authors:** Takafumi Osaka, Yoshitaka Hashimoto, Takuro Okamura, Takuya Fukuda, Masahiro Yamazaki, Masahide Hamaguchi, Michiaki Fukui

**Affiliations:** 1Department of Endocrinology and Metabolism, Kyoto Prefectural University of Medicine, Graduate School of Medical Science, Kyoto 602-8566, Japan; tak-1314@koto.kpu-m.ac.jp (T.O.); y-hashi@koto.kpu-m.ac.jp (Y.H.); d04sm012@koto.kpu-m.ac.jp (T.O.); takuya07@koto.kpu-m.ac.jp (T.F.); masahiro@koto.kpu-m.ac.jp (M.Y.); mhama@koto.kpu-m.ac.jp (M.H.); 2Department of Endocrinology and Diabetology, Ayabe City Hospital, Ayabe 623-0011, Japan

**Keywords:** liver stiffness, fatty liver, skeletal muscle mass, body composition, type 2 diabetes

## Abstract

Body weight reduction leads to improvement of nonalcoholic fatty liver disease (NAFLD), but the contributions of body composition modification on its improvement have not been clarified yet. We performed a retrospective cohort study in a Japanese university hospital to clarify the effect of body fat reduction on the improvement of hepatic stiffness as well as hepatic steatosis. The skeletal muscle mass index (SMI, kg/m^2^), fat to muscle mass ratio, and the change in fat to muscle mass ratio after 1 year from baseline were calculated. Controlled attenuation parameter (CAP, dB/m) and liver stiffness measurement (LSM, kPa) were evaluated by elastography. Primary outcome was set as the association of the change of fat to muscle mass ratio after 1 year from baseline with the change of liver stiffness measurement. One hundred and seventeen patients (59 men and 58 women) completed the study. The average age was 63.5 years, and baseline CAP and LSM were 273.4 ± 53.5 dB/m and 6.3 ± 3.4 kPa, respectively. After 1 year, body mass index (BMI), SMI, and LSM decreased. Multiple regression analyses demonstrated that change in fat to muscle mass ratio was associated with the change in CAP (*ß* = 0.38, *p* < 0.001) or LSM (*ß* = 0.21, *p* = 0.026). The reduction of fat to muscle mass ratio was associated with improvement in liver stiffness, but the reduction of BMI was not.

## 1. Introduction

Nonalcoholic fatty liver disease (NAFLD) is a frequent cause of liver-related morbidity and mortality [1,2]. NAFLD comprises a range of conditions from simple steatosis to nonalcoholic steatohepatitis [3]. Approximately 60% of type 2 diabetes patients have NAFLD [4] and a part of patients progress from NAFLD to liver cirrhosis and hepatocellular carcinoma [5]. It is important to treat NAFLD in diabetes patients to prevent further progression of liver disease and to improve diabetes [6].

It is well known that body weight reduction leads to improvement of NAFLD [7]. In addition to that, the modification in body composition contributes to preventing life-threatening diseases [8]. Loss of body weight includes reductions of both body fat and muscle mass [9,10]. Sarcopenia, characterized by a decline in muscle and low muscle strength, affects clinical outcomes, including risk of cardiovascular disease and infection [11,12]. In addition, high body fat and low skeletal muscle are associated with liver steatosis and fibrosis [4,13,14,15]. Thus, there is a possibility that not the body weight reduction, but the body fat reduction and the skeletal muscle gain is important for NAFLD. However, the effect of reduction of body fat and skeletal muscle on liver stiffness as well as steatosis has not been clarified yet.

This study investigated the relationships of the reduction of body fat with the improvement of liver stiffness as well as steatosis in a cohort of type 2 diabetes patients with NAFLD.

## 2. Methods

### 2.1. Study Population and Design

We performed a multipurpose cohort study in a single center Japanese university hospital. The aim of the study was to obtain the clinical features of subjects with NAFLD and to prevent the progression of nonalcoholic steatohepatitis (NASH). For the purpose, we analyzed the clinical data registered in the study and found clinical features associated with the progression of NASH and found clinical features, by which we can predict a high-risk group of NASH. The study was approved Kyoto Prefectural University of Medicine, Clinical Research Review Board at 2 March 2015. The project identification code is ERB-C-297. The study protocol registered at UMIN (UMIN000020303). We previously reported two cross-sectional studies in which study periods were set as January 2015 to September 2015, or January 2015 to January 2016, respectively [4,13].

In this study, to clarify the association of the change of body composition change with the change of liver steatosis or stiffness, the participants needed two or more elastography and bioelectrical impedance body composition evaluations. The written informed consent was obtained from all the participants.

### 2.2. Sample Size and Periods for the Recruitments

The sample size was not estimated a priori, because this is an exploratory study. In this study, the inclusion period was set between March 2015 to August 2016 and based on practical reason. We included the patients who underwent both elastography and bioelectrical impedance body composition evaluations. We excluded the patients without NAFLD, the definition is described below. Exclusion criteria was set as no data of lever stiffness measurement, or no data of bio impedance analysis. After initial registration, the patients, with two or more elastography and bioelectrical impedance body composition evaluations, were extracted (Figure 1).

### 2.3. Definition for NAFLD

In this study, fatty liver was diagnosed by the findings of abdominal ultrasonography performed by trained technicians. The participants with liver contrast and liver brightness among the four known criteria (hepatorenal echo contrast, liver brightness, deep attenuation, and vascular blurring) were diagnosed as having fatty liver [16]. Patients were asked about the duration of diabetes, exercise habits, smoking status, the amount and types of alcoholic beverages consumed weekly during the past month. Men with a mean weekly ethanol consumption of more than 210 g, women who consumed more than 140 g/week, and patients with known liver disease were excluded [17]. Regarding known liver disease, participants who tested positive for hepatitis B antigen or hepatitis C antibody and those who reported a history of known liver disease, including viral, genetic, autoimmune, and drug-induced liver disease, were also excluded [18]. No patients had cancer or autoimmune diseases. The patients who had fatty liver and who did not have exclusion criteria were defined as having NAFLD.

### 2.4. Biochemical Assays

Blood samples were drawn from the antecubital vein of seated patients after 8 h of fasting. Serum aspartate aminotransferase (AST), alanine aminotransferase (ALT), and gamma-glutamyl transferase (GGT) were assayed by standard enzymatic methods. Hemoglobin A1c (HbA1c) was assayed by high-performance liquid chromatography and reported in National Glycohemoglobin Standardization Program units. The fibrosis (FIB) 4 index was calculated as: age × AST/(platelet count × rout ALT) [19].

### 2.5. Measurement of Body Composition

Body composition was measured with a segmental multifrequency bioelectrical impedance body composition analyzer (Inbody 720; Inbody Japan Inc., Tokyo, Japan). Multifrequency impedance body composition measurements were correlated with dual energy X-ray absorptiometry and were validated [20,21]. Body weight (kg), body fat mass (BFM, kg), and appendicular skeletal muscle mass (kg) were measured. The body fat percentage was calculated as ((BFM/body weight) × 100). The skeletal muscle mass index (SMI) was the appendicular skeletal muscle mass divided by the height in meters squared [22]. Body mass index (BMI) was calculated as body weight divided by height in meters squared (kg/m^2^). The fat to muscle mass ratio was the BFM divided by the appendicular skeletal muscle mass [23]. The change in fat to muscle mass ratio was the difference of the value at 1 year and the baseline value.

### 2.6. The Controlled Attenuation Parameter (CAP) and Liver Stiffness Measurement (LSM)

CAP and LSM were determined by FibroScan 502 (Echosens, Paris, France) [24]. The measurements were made on patients in the supine position using B-mode ultrasound. Liver elastometry was performed with a range of probe pressures at an intercostal area. The measurement of CAP and LSM has been described elsewhere [25,26]. The rate of change in CAP or LSM was the difference of the second and baseline measurements divided by the baseline value. The cutoff point for liver stiffness values of fibrosis categories were 7.6 kPa for extensive liver stiffness [26]. In this study, we also divided the patients into four groups: group 1, the patients who were normal liver stiffness both at baseline and follow-up examinations; group 2, the patients who changed from normal to extensive liver stiffness; group 3, the patients who were extensive liver stiffness both at baseline and follow-up examinations; and group 4, the patients who change from extensive to normal liver stiffness.

### 2.7. Statistical Analysis and Missing Values

Statistical analysis was performed with JMP version 12.0 (SAS Institute Inc., Cary, NC, USA) or SPSS version 25.0 (SPSS Inc., Chicago IL USA). Missing clinical data were handled by multiple imputation in R multivariate imputation by chained equation (MICE) package repeated ten times to properly account for variability due to unknown values. Continuous variables were reported as means ± standard deviation (SD). Categorical variables were reported as numbers and percentages. The paired *t*-test (continuous variables) and Wilcoxon signed-rank test (categorical variables) were used to identify the significance of differences in baseline and 1-year values. The difference among groups were evaluated by one-way ANOVA and Tukey–Kramer HSD test. The association of CAP or LSM and metabolic variables were determined by Spearman’s rank correlation coefficient. Regression analysis was used to evaluate the effects of various factors on the rate of change in CAP or LSM. Age, sex, duration of diabetes, smoking status, exercise, HbA1c, BMI, start of sodium glucose cotransporter 2 inhibitor (SGLT2i) or glucagon-like peptide-1 receptor agonist (GLP-1RA), change in BMI, and change in fat to muscle mass ratio were independent variables in multiple regression analysis.

## 3. Results

A cohort of 246 study patients with type 2 diabetes mellitus, 140 men and 106 women, were evaluated. Seventy-two men and 31 women were excluded from the analysis (Figure 1). Of the remaining 143 patients (68 men and 75 women), 26 (nine men and 17 women) did not receive a 1-year follow-up examination. We have analyzed the difference of anthropometric indices and laboratory parameters in the patients with and without performing follow-up (Supplementary Materials). There was no difference between the patients with and without performing follow-up.

A total of 117 patients (59 men and 58 women) were included in the final analysis. The clinical characteristics of the 117 patients are shown in Table 1. Mean age was 63.5 ± 12.2 years, the BMI was 25.4 ± 4.4 kg/m^2^, the SMI was 7.1 ± 1.2 kg/m^2^, the body fat percentage was 31.1% ± 8.2%, the HbA1c was 7.5% ± 1.1%, the AST was 29.8 ± 18.6 IU/L, the ALT was 34.9 ± 30.0 IU/L, Fib-4 index 1.67 ± 0.84, the CAP was 273.4 ± 53.5 dB/m, and the LSM 6.3 ± 3.4 kPa. At the 1-year follow-up, the mean BMI, SMI, HbA1c, and LSM values significantly decreased. The platelet counts significantly increased. The AST, ALT, and CAP were no apparent change. The use of metformin, SGLT2i, and GLP-1RA had increased, and the use of sulfonylurea decreased.

In this study, 83 (70.9%) patients were normal liver stiffness both at baseline and follow-up examinations (group 1); 7 (6.0%) patients changed from normal to extensive liver stiffness (group 2); 10 (8.5%) patients were extensive liver stiffness both at baseline and follow-up examinations (group 3); and 17 (14.5%) patients changed from extensive to normal liver stiffness (group 4). There was no difference of the change in BMI, the difference of the value at 1 year and the baseline value, among the groups (−0.23 (0.96) in group 1, −0.64 (0.68) in group 2, −0.75 (0.85) in group 3, and 0.17 (0.71) kg/m^2^ in group 4, (*p* = 0.069) (Figure 2). The change in fat-to-muscle ratio was -4.03 (30.7)% in group 1, −40.6 (12.4)% in group 2, −14.4 (34.6)% in group 3 and 117.2 (180.3)% in group 4 (*p* < 0.001) and that of group 3 was higher than that of group 3 (*p* < 0.001), that of group 2 (*p* < 0.001) and group 1 (*p* < 0.001) (Figure 2).

Group 1, the patients who were normal liver stiffness both at baseline and follow-up examinations; Group 2, the patients who changed from normal to extensive liver stiffness; Group 3, the patients who were extensive liver stiffness both at baseline and follow-up examinations; and Group 4, the patients who changed from extensive to normal liver stiffness.

The associations of CAP or LSM and baseline metabolic variables are shown in Table 2. The BMI (*r* = 0.56, *p* < 0.001), appendicular skeletal muscle mass (*r* = 0.31, *p* < 0.001), SMI (*r* = 0.35, *p* < 0.001), body fat percentage (*r* = 0.41, *p* < 0.001), fat-to-muscle ratio (*r* = 0.37, *p* < 0.001), or Fib-4 index (*r* = −0.33, *p* < 0.001) was associated with CAP. The BMI (*r* = 0.39, *p* < 0.001), SMI (*r* = 0.24, *p* = 0.008), body fat percentage (*r* = 0.24, *p* = 0.010), fat-to-muscle ratio (*r* = 0.22, *p* = 0.018) was associated with LSM, whereas Fib-4 index (*r* = 11, *p* = 0.233) was not associated with LSM.

Multiple regression analyses (Table 3) demonstrated that change in fat-to-muscle mass ratio was associated with the rate of change in CAP (*ß* = 0.38, *p* < 0.001) or rate of change in LSM (*ß* = 0.21, *p* = 0.026). Change in BMI was associated with the rate of change in CAP (*ß* = 0.38, *p* < 0.001), but not with the rate of change in LSM (*ß* = 0.15, *p* = 0.123).

## 4. Discussion

This study investigated the association of metabolic or physical variables and improvement of liver stiffness as well as steatosis in type 2 diabetes patients with NAFLD. We revealed that the reduction of fat-to-muscle mass ratio was associated with improvement of liver stiffness but change in BMI was not associated with improvement of liver stiffness.

Previous studies reported that reduction of body weight led to improvement of liver steatosis [7], and that low muscle mass has been associated with liver steatosis [4] or stiffness [13,26]. Reduction of body weight has been associated with the low incidence of NAFLD [10]. However, it is not body weight alone, but body composition that influences the progression to NAFLD [17].

The possible mechanisms of change in fat-to-muscle mass ratio and improved liver steatosis or stiffness are described below. Increased insulin resistance in muscle, which is associated with a low skeletal muscle percentage, leads to increased glucose uptake by the liver and the relatively low insulin resistance in liver leads to steatosis [13,27]. Myokines may also affect liver steatosis and stiffness. Muscle-derived interleukin (IL)-6 induces transient signal transducer and activator of transcription 3 activation [28] and a proinflammatory response [27] that protects against liver fibrosis [29]. Irisin, which is an exercise-induced myokine, converts white adipose to brown adipose tissue [30] and leptin, which is secreted by fat mass, affects liver fibrosis through chronic inflammation [31]. In addition, it has been reported that a reduction of visceral fat stores is an essential treatment target to manage NAFLD [32,33]. The reduction in visceral fat accumulation has association with improvements in insulin resistance and systemic chronic low-grade inflammation, and it has also decreased the risk of cardiovascular disease. East Asian people, including the Japanese, have a higher percentage of body fat than Caucasians [34]; thus, the reduction of body fat is an important treatment target. The above findings support our results.

Both skeletal muscle and body fat mass showed a significant reduction during the 12 months of follow-up. The reduction of body fat without skeletal muscle usually needs high protein nutrition therapy and high load resistance training [27]. However, neither diet therapy nor exercise therapy have been changed during this one-year period.

Many patients in this study cohort had recently started taking SGLT2i or GLP-1 RA to treat NASH or NAFLD with type 2 diabetes [35]. However, the use of SGLT2i and GLP-1 RA was not associated with improvement of liver steatosis. A recent study reported the effect on liver steatosis by SGLT2i [36,37,38], but, the change in CAP was 0.27 ± 27.9 in patients without using SGLT2i or GLP-1 RA vs. 1.26 ± 25.1 in patients with using SGLT2i, *p* = 0.599 (Kruskal–Wallis test). A recent study reported the effect on liver stiffness by GLP-1RA [37,38,39], but, the change in liver stiffness was 0.27 ± 27.9 in patients without using SGLT2i or GLP-1 RA and −0.67 ± 28.6 in patients with GLP-1 RA, *p* = 0.789 (Kruskal–Wallis test). The study design might be influenced by the difference. In other words, the design of the reported studies are randomized control studies, but this study design is a retrospective investigation study.

The study limitations include not evaluating the liver steatosis or fibrosis by liver biopsy, even though transient elastography has been validated for diagnosis of liver steatosis [40] or stiffness [41,42]. Dual energy X-ray absorptiometry is the gold standard assay for evaluating the skeletal muscle volume. However, a multifrequency impedance body composition analyzer was used. The values obtained by this method are well correlated with those obtained, which has a good correlation with X-ray absorptiometry, and it has been validated [21]. As only Japanese patients were included, the results may not be applicable in other populations. A high dropout rate might be the cause of a selection bias. Ultrasonographic semi-quantitative indices which predict liver histology and correlate with metabolic parameters were not used. In addition, owing to too few individuals being recruited, separate analysis by sex was not performed, therefore neglecting recent evidence of sex differences and sex dimorphism in NAFLD. The patients with marginal values of CAP and LSM may lead to either under- or overestimation of the results, although the definition for marginal values of CAP and LSM has not been established.

## 5. Conclusions

The reduction of fat-to-muscle mass ratio was associated with improvement of liver stiffness but change in BMI was not. These results suggested that to improve liver steatosis and stiffness of the patients with type 2 diabetes, the physicians should focus on not only BMI, but the body compositions, such as body fat and muscle mass.

## Figures and Tables

**Figure 1 jcm-08-02175-f001:**
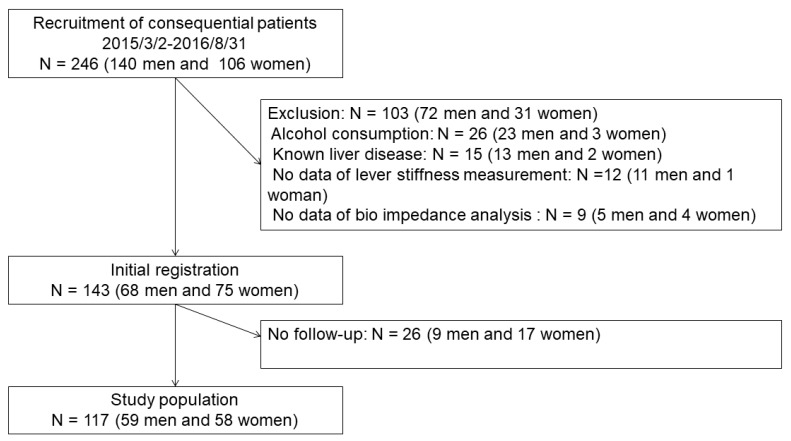
Patient inclusion, exclusion, and disposition.

**Figure 2 jcm-08-02175-f002:**
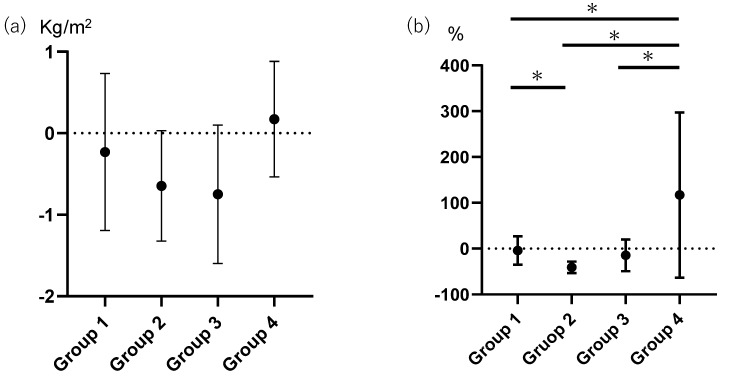
The difference of the change in BMI or fat-to-muscle ratio among the live stiffness status. (**a**) The difference of the change in BMI or among the live stiffness status. There was no difference of the change in BMI among the groups (*p* = 0.069, by one-way ANOVA). (**b**) The difference of the change in fat-to-muscle ratio among the live stiffness status (*p* < 0.001, by one-way ANOVA). The change in fat-to-muscle ratio of group 3 was higher than that of group 2 (*p* < 0.001, by Tukey–Kramer HSD test), that of group 1 (*p* < 0.001, by Tukey–Kramer HSD test) and group 0 (*p* < 0.001, by Tukey–Kramer HSD test). The change in fat-to-muscle ratio of group 0 was higher than that of group 1 (*p* = 0.036, by Tukey–Kramer HSD test). * *p* < 0.05.

**Table 1 jcm-08-02175-t001:** Characteristic of study participants.

	Baseline			After One Year			
	All	Men	Women	All	Men	Women	*p*
N	117	59	58				
Age, year	63.5 (12.2)	64.1 (11.5)	63.0 (12.9)				
Duration of diabetes, years	12.0 (9.6)	12.5 (9.5)	11.4 (9.7)				
Regular exercise	48 (41.0)	25 (42.4)	23 (39.7)				
Smoking Status							
Never smoker	56 (47.9)	15 (25.4)	41 (70.7)				
Former smoker	46 (39.3)	33 (55.9)	13 (22.4)				
Current smoker	15 (12.8)	11 (18.6)	4 (6.9)				
Body mass index, kg/m^2^	25.4 (4.4)	25.7 (4.1)	25.0 (4.7)	25.1 (4.2)	25.3 (3.8)	24.8 (4.6)	<0.001
Appendicular skeletal muscle mass, kg	19.0 (4.8)	22.0 (3.8)	15.8 (3.4)	18.7 (4.7)	21.8 (3.7)	15.5 (3.4)	<0.001
Skeletal muscle index, kg/m^2^	7.1 (1.2)	7.8 (1.0)	6.5 (0.9)	7.0 (1.2)	7.7 (0.9)	6.4 (0.9)	<0.001
Body fat mass, kg	21.2 (8.2)	20.7 (7.4)	21.8 (8.9)	20.8 (8.0)	20.1 (6.9)	21.5 (8.9)	0.019
Body fat percentage, %	31.1 (8.2)	28.0 (6.5)	34.4 (8.5)	31.0 (8.0)	27.7 (6.2)	34.4 (8.2)	0.440
Fat-to-muscle ratio	0.9 (0.3)	0.7 (0.2)	1.0 (0.4)	0.9 (0.3)	0.7 (0.2)	1.0 (0.4)	0.478
Systolic blood pressure, mmHg	134.0 (19.8)	132.2 (18.7)	135.8 (20.8)	135.9 (17.0)	134.6 (15.6)	137.1 (18.4)	0.305
Diastolic blood pressure, mmHg	79.1 (11.6)	80.5 (11.1)	77.6 (12.1)	78.4 (10.3)	79.7 (9.8)	77.1 (10.7)	0.515
Platelet count, × 10^9^/L	218.6 (53.9)	209.0 (43.9)	228.5 (61.3)	226.8 (57.0)	220.8 (50.3)	232.9 (63.0)	0.010
Hemoglobin A1c, %	7.5 (1.1)	7.7 (1.2)	7.3 (1.0)	7.2 (1.2)	7.4 (1.2)	7.0 (1.0)	<0.001
Hemoglobin A1c, mmol/L	58.4 (12.4)	60.8 (13.4)	55.9 (11.0)	54.9 (12.6)	57.0 (13.4)	52.8 (11.4)	<0.001
Aspartate aminotransferase, IU/L	29.8 (18.6)	32.2 (22.6)	27.3 (13.2)	27.7 (15.5)	29.8 (16.1)	25.7 (14.8)	0.193
Alanine aminotransferase, IU/L	34.9 (30.0)	39.5 (36.6)	30.2 (20.6)	31.2 (24.9)	35.3 (29.0)	26.9 (19.2)	0.079
Gamma-glutamyl transferase, IU/L	47.1 (46.4)	56.6 (54.4)	37.4 (34.4)	42.9 (50.6)	52.7 (62.8)	32.9 (31.6)	0.113
Ferritin, ng/mL	126.1 (116.6)	154.6 (137.2)	97.2 (82.6)	114.6 (108.3)	143.2 (131.6)	85.5 (67.4)	0.120
Type 4 collagen 7 S, ng/mL	4.8 (1.0)	4.9 (1.2)	4.8 (0.8)	5.0 (1.2)	5.0 (1.3)	5.0 (1.1)	0.182
Hyaluronic acid, ng/mL	85.0 (134.2)	87.9 (173.8)	82.1 (76.9)	75.1 (71.5)	72.1 (76.1)	78.1 (66.9)	0.337
Fib-4 index	1.67 (0.84)	1.78 (0.95)	1.55 (0.71)	1.58 (0.74)	1.65 (0.8)	1.5 (0.67)	0.053
Controlled attenuation parameter, dB/m	273.4 (53.5)	279.1 (50.9)	267.6 (55.8)	269.7 (70.8)	273.7 (71.4)	265.6 (70.6)	0.510
Liver stiffness measurement, kPa	6.3 (3.4)	7.0 (4.2)	5.5 (2.1)	5.6 (2.5)	5.8 (3.0)	4.8 (1.7)	0.040
Extensive liver stiffness	27 (23.1)	17 (28.8)	10 (17.2)	17 (14.5)	12 (20.3)	5 (8.6)	<0.001
Sulfonylurea	42 (35.9)	23 (39.0)	19 (32.8)	35 (30.0)	19 (32.2)	16 (27.6)	0.034
Dipeptidyl peptidase-4 inhibitors	34 (29.1)	22 (37.3)	12 (20.7)	38 (32.4)	20 (33.9)	18 (31.0)	0.396
Glinides	8 (6.8)	3 (5.1)	5 (8.6)	7 (6.0)	1 (1.7)	6 (10.3)	0.657
Metformin	54 (46.2)	29 (49.2)	25 (43.1)	60 (51.3)	34 (57.6)	26 (44.8)	0.033
Thiazolidine	5 (4.3)	3 (5.1)	2 (3.4)	3 (2.6)	2 (3.4)	1 (1.7)	0.158
α-glucosidase inhibitors	14 (12.0)	8 (13.6)	6 (10.3)	9 (7.7)	7 (11.9)	2 (3.4)	0.096
Sodium-glucose cotransporter-2 inhibitors	2 (1.7)	2 (3.4)	0 (0)	25 (21.4)	10 (16.9)	15 (25.9)	<0.001
Insulin	27 (23.1)	17 (28.8)	10 (17.2)	25 (21.4)	17 (28.8)	8 (13.8)	0.319
Glucagon-like peptide-1	5 (4.3)	1 (1.7)	4 (6.9)	30 (25.6)	16 (27.1)	14 (24.1)	<0.001

Continuous variables were presented as the mean (standard deviation; SD) and categorical variables were presented as number (percentage). The paired t test (continuous variable) and Wilcoxon signed-rank test (categorical variable) were performed to identify the statistical differences between baseline and one year after.

**Table 2 jcm-08-02175-t002:** Simple correlation between controlled attenuation parameter or liver stiffness measurement and metabolic parameters.

	Controlled Attenuation Parameter		Liver Stiffness Measurement	
	*r*	*p*	*r*	*p*
Age	−0.39	<0.001	−0.14	0.135
Body mass index	0.56	<0.001	0.39	<0.001
Appendicular skeletal muscle mass	0.31	<0.001	0.18	0.055
Skeletal muscle index	0.35	<0.001	0.24	0.008
Body fat percentage	0.41	<0.001	0.24	0.01
Fat-to-muscle ratio	0.37	<0.001	0.22	0.018
Hemoglobin A1c	0.29	<0.001	0.14	0.137
Aspartate aminotransferase	0.28	0.003	0.54	<0.001
Alanine aminotransferase	0.48	<0.001	0.52	<0.001
Gamma-glutamyl transferase	0.32	<0.001	0.47	<0.001
Fib-4 index	−0.33	<0.001	0.11	0.233

**Table 3 jcm-08-02175-t003:** Multiple regression analysis of the effects of various factors on rate of change in controlled attenuation parameter or liver stiffness measurement.

	Rate of Change in Controlled Attenuation Parameter			
	*β*	*p*	*β*	*p*
Age, year	−0.10	0.383	−0.14	0.215
Men	0.07	0.472	0.07	0.440
Smoking	−0.12	0.181	−0.13	0.160
Exercise	0.01	0.890	0.05	0.616
Duration of diabetes, year	0.03	0.795	0.0004	0.997
Hemoglobin A1c, %	0.05	0.584	0.10	0.285
Body mass index, kg/m^2^	0.26	0.021	0.19	0.080
Sodium glucose cotransporter 2 inhibitor	−0.09	0.422	−0.11	0.289
Glucagon-like peptide-1	−0.03	0.709	−0.05	0.573
Change in body mass index, kg/m^2^	0.38	<0.001	−	−
Change in fat-to-muscle ratio, Δ1 incremental	−	−	0.38	<0.001
	**Rate of Change in Liver Stiffness Measurement**			
	***β***	***p***	***β***	***p***
Age, year	0.02	0.834	0.02	0.877
Men	0.08	0.451	0.07	0.459
Smoking	−0.06	0.527	−0.07	0.473
Exercise	0.07	0.485	0.09	0.376
Duration of diabetes, year	0.15	0.179	0.14	0.204
Hemoglobin A1c, %	−0.15	0.153	−0.13	0.204
Body mass index, kg/m^2^	0.04	0.763	0.01	0.900
Sodium glucose cotransporter 2 inhibitor	−0.05	0.655	−0.06	0.608
Glucagon-like peptide-1	−0.08	0.439	−0.08	0.409
Change in body mass index, kg/m^2^	0.15	0.123	−	−
Change in fat-to-muscle ratio, Δ1 incremental	−	−	0.21	0.026

Smoking was used non- and past smoking as a reference.

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
