# Peer review of "Reduction of Fat to Muscle Mass Ratio Is Associated with Improvement of Liver Stiffness in Diabetic Patients with Non-Alcoholic Fatty Liver Disease"

_jcm, 2019, doi:10.3390/jcm8122175_

Round 1
Reviewer 1 Report
The study by Masahide Hamaguchi et al entitled “Reduction of fat to muscle mass ratio is associated with improvement of liver stiffness in diabetic patients with non-alcoholic fatty liver disease” has been reviewed. It is a retrospective cohort study in a Japanese University Hospital with the aim clarify the effect of body fat reduction on the improvement of hepatic stiffness as well as hepatic steatosis and showing that the reduction of fat to muscle mass ratio was associated with improvement in liver stiffness, but the reduction of BMI was not. The manuscript is clear and well designed; nonetheless the discussion is not entirely and clearly debated regarding the many patients recently started taking SGLT2i or GLP-1 RA to treat NASH or NAFLD with type 2 diabetes and the described possible mechanisms of change in fat to muscle mass ratio and improved liver steatosis, I would like to suggest to the authors to improve the discussion section. I am positive about this study, but I would urge the authors to accommodate the recommended suggestion, as I believe the results can be of benefit to the research community.
Author Response
The manuscript is clear and well designed; nonetheless the discussion is not entirely and clearly debated regarding the many patients recently started taking SGLT2i or GLP-1 RA to treat NASH or NAFLD with type 2 diabetes and the described possible mechanisms of change in fat to muscle mass ratio and improved liver steatosis, I would like to suggest to the authors to improve the discussion section. I am positive about this study, but I would urge the authors to accommodate the recommended suggestion, as I believe the results can be of benefit to the research community.
Response
Thank you for your suggestion. According to your suggestion, we have added the possible mechanisms of change in fat to muscle mass ratio and improved liver steatosis in the Discussion section, described as below.
“In addition, it has been reported that a reduction of visceral fat stores is essential treatment target for manage NAFLD.[a,b] The reduction in visceral fat accumulation has association with improvements in insulin resistance and systemic chronic low-grade inflammation, and it has also decreased the risk of cardiovascular disease. East Asian people, including the Japanese, have a higher percentage of body fat than Caucasians;[c] thus the reduction of body fat is important treatment target.”
a. Bouchi R, Nakano Y, Fukuda T, et al. Reduction of visceral fat by liraglutide is associated with ameliorations of hepatic steatosis, albuminuria, and micro-inflammation in type 2 diabetic patients with insulin treatment: a randomized control trial. Endocr J. 2017;64:269–81.
b.Houghton D, Thoma C, Hallsworth K, et al. Exercise reduces liver lipids and visceral adiposity in patients with nonalcoholic steatohepatitis in a randomized controlled Trial. Clin Gastroenterol Hepatol. 2017;15:96–102.
c.WHO Expert Consultation. Appropriate body-mass index for Asian populations and its implications for policy and intervention strategies. Lancet. 2004;363:157–63.
Reviewer 2 Report
The authors of the study aimed to find the factors related to body mass and body composition, influencing the live stiffness in patients with diabetes and NAFLD. I have some major and minor concerns, as follows:
The authors write, that change in BMI or fat to muscle ratio was associated with decrease in CAP and LSM, however the results do not clearly show it; I would like to see exactly in how many patients and with what BMI/fat to muscle ratio change did the liver stiffness increase, decrease or stay constant; a plot might be helpful in visualizing the observed changes I think that the authors should provide some more information about the baseline liver stiffness measurements, in particular how many patients did have extensive or normal liver stiffness; were there more patients with normal liver stiffness, or with signs of liver fibrosis in CAP/LSM measurement? I am concerned about the patients with marginal values of CAP and LSM, as an excessive number of patients in either may lead to either under- or overestimation of the results. Authors write, that one one the exclusion criteria was known liver disease, but NAFLD is a liver disease itself, please specify English editing is needed, some of the paragraphs are well-written, but some need revisions, for example paragraphs 2-1 and 2-2 in the methods section.
Author Response
I would like to see exactly in how many patients and with what BMI/fat to muscle ratio change did the liver stiffness increase, decrease or stay constant; a plot might be helpful in visualizing the observed changes.
I think that the authors should provide some more information about the baseline liver stiffness measurements, in particular how many patients did have extensive or normal liver stiffness;
Were there more patients with normal liver stiffness, or with signs of liver fibrosis in CAP/LSM measurement?
Response
Thank you for your comment. As you say, a plot might be helpful in visualizing the observed changes. When we use the cutoff point for liver stiffness values of fibrosis categories were 7.6 kPa for extensive liver stiffness, 27 (23.1%) patients were extensive liver stiffness at baseline examination.
In this study, 83 (70.9%) patients were normal liver stiffness both at baseline and follow-up examinations (group 1); 7 (6.0%) patients change from normal to extensive liver stiffness (group 2); 10 (8.5%) patients were extensive liver stiffness both at baseline and follow-up examinations (group 3); and 17 (14.5%) patients change from extensive to normal liver stiffness (group 4).
There was no difference of the change in BMI, the difference of the value at 1 year and the baseline value, among the groups (-0.23 (0.96) in group 1, -0.64 (0.68) in group 2, -0.75 (0.85) in group 3 and 0.17 (0.71) kg/m2 in group 4, p = 0.069 by one-way ANOVA).
The change in fat to muscle ratio was 0.004 (0.089) in group 1, -0.06 (0.08) in group 2, -0.03 (0.08) in group 3 and 0.04 (0.05) in group 4 (p = 0.019 by one-way ANOVA) and that of group 3 was higher than that of group 1 (p = 0.038, by Tukey-Kramert HSD test) and group 1 (p = 0.048, by Tukey-Kramert HSD test).
According to your comment, we have added new Figure 2 and these points in the Methods and Results sections described as below.
Methods section
“The cutoff point for liver stiffness values of fibrosis categories were 7.6 kPa for extensive liver stiffness.[d] In this study, we also divided the patients into four groups: group 1, the patients who were normal liver stiffness both at baseline and follow-up examinations; group 2, the patients who change from normal to extensive liver stiffness; group 3, the patients who were extensive liver stiffness both at baseline and follow-up examinations; and group 4, the patients who change from extensive to normal liver stiffness.”
“The difference among groups were evaluated by one-way ANOVA and Tukey-Kramert HSD test.”
Results section
“In this study, 83 (70.9%) patients were normal liver stiffness both at baseline and follow-up examinations (group 1); 7 (6.0%) patients change from normal to extensive liver stiffness (group 2); 10 (8.5%) patients were extensive liver stiffness both at baseline and follow-up examinations (group 3); and 17 (14.5%) patients change from extensive to normal liver stiffness (group 4). There was no difference of the change in BMI, the difference of the value at 1 year and the baseline value, among the groups (-0.23 (0.96) in group 1, -0.64 (0.68) in group 2, -0.75 (0.85) in group 3 and 0.17 (0.71) kg/m2 in group 4, p = 0.069) (Figure 2). The change in fat to muscle ratio was 0.004 (0.089) in group 1, -0.06 (0.08) in group 2, -0.03 (0.08) in group 3 and 0.04 (0.05) in group 4 (p = 0.019) and that of group 3 was higher than that of group 1 (p = 0.038) and group 1 (p = 0.048) (Figure 2).”
d. Saito H, Tada S, Nakamoto N, Kitamura K, Horikawa H, Kurita S, Saito Y, Iwai H, Ishii H. Efficacy of non-invasive elastometry on staging of hepatic fibrosis. Hepatol Res. 2004 Jun;29(2):97-103.
Comment 4; I am concerned about the patients with marginal values of CAP and LSM, as an excessive number of patients in either may lead to either under- or overestimation of the results.
Response
Thank you for your comment. As you say, the patients with marginal values of CAP and LSM may lead to either under- or overestimation of the results. We have inserted your concern in the limitation section.
Comment 5; Authors write, that one the exclusion criteria was known liver disease, but NAFLD is a liver disease itself, please specify
Response
Thank you for your comment. According to your comment, we have added the more precious exclusion criteria in the Methods section described as below.
“Regarding known liver disease, participants who tested positive for hepatitis B antigen or hepatitis C antibody and those who reported a history of known liver disease, including viral, genetic, autoimmune, and drug-induced liver disease, were also excluded.[e]”
e. McCullough AJ. The clinical features, diagnosis and natural history of nonalcoholic fatty liver disease. Clin Liver Dis. 2004 Aug;8(3):521-33
Comment 5; English editing is needed, some of the paragraphs are well-written, but some need revisions, for example paragraphs 2-1 and 2-2 in the methods section.
Response
Thank you for your comment. According to your comment, we have editing by native English speaker.
Reviewer 3 Report
GENERAL COMMENT
The manuscript is quite inaccurately written (I give several specific examples below of wrong, repetitive and unclear sentences). The criteria for diagnosing NAFLD are wrong and these authors confuse steatosis (i.e. fat in the liver) with NAFLD (i.e. steatosis + exclusion of competing causes of steatosis). In addition they do not seem to use a specific laboratory protocol to rule out such competing causes of CLD. The introduction is too short and the rationale is not well described. References are not invariably well taken. Some conclusions are too strong and must be softened owing to the many limitations of the study design which should be honestly admitted and discussed. Editing of English is recommended.
SPECIFIC COMMENTS
Approximately 60% of diabetes patients have NAFLD--> please, specify what type of diabetes
References
Ref 5 in supporting the statement “many patients progress from NAFLD to liver cirrhosis and hepatocellular carcinoma” is outdated and non-specific.
Line 35 “ NAFLD in diabetes patients to prevent further development of liver disease and to improve” → rather than “further development I would say “the progression” or “worsening” .
Lines 37-40 do not match well with lines 41-44 and should probably be omitted. Conversely Lines 41-44 are well focussed on the problem but do not say and instead must say) what – based on literature studies - these authors expected to find and why. In other words the rationale should be better elucidated.
Throughout the manuscript the word non-alcoholic vs. nonalcoholic is inconsistently written. Please only one spelling (nonalcoholic).
Lines 52 and 54 contain the same abbreviation twice (NASH). One is enough: the first
“We previously performed a two cross-sectional study in which study period was set as January 2015 57 to September 2015, or January 2015 to January 2016, respectively” sentence unclear. Is this one study or 2 studies ?
Line 59 “ In this study, the research question as to clarify “ IS ?
Line 61 – Consequential → Consecutive ?
Lines 63-64 “The patients were not involved in the recruitment to and conduct of the study.” Is this required by Ethycal Commette ? This is a quite uncommon statement.
Line 68 The sample size was not estimated priori → A PRIORI
The submission has many repetitions which slow down reading for example “this study is exploratory prospective study “
Please, reword the follwing more concisely without any repetitions “In this study, the research question as to clarify the association of the change of body composition change with liver stiffness as well as liver steatosis was set after collecting measurements, and the study design was a retrospective analysis in a multipurpose prospective cohort study. “
“because this study is exploratory prospective study. In this retrospective cohort study, inclusion period “ In the word study is reated three times and study turns from prospective to retrospective (?)
Again, things are repeated twice here “Exclusion criteria was set as alcohol consumption, known liver disease, [7] no data 70 of lever stiffness measurement, or no data of bio impedance analysis. Men with a mean weekly ethanol consumption of more than 210 g, women who consumed more than 140 g/week, and patients with known liver disease were excluded according to the criterion of alcohol consumption”
English is poor “study population was consisted with” “Researchers who are specialized in diabetology are conducted in clinical practice in this cohort “
Lines 86-87 “Men with a mean weekly ethanol consumption of more than 210 g, women who consumed more than 140 g/week, and patients with known liver disease were excluded.” exactly duplicates lines 71 and 72.
Throughut th emanuscript “Known liver disease”--> What does this mean ? Based on medical hstory ? No laboratory test to rule out competing etiologies. No questions on potentially steatogenic drug treatment ?
“NAFLD was diagnosed by abdominal ultrasonography” This is conceptually wrong. Ultrasonography cannot see NAFLD although it can appreciate the presence of STEATOSIS. IF, as I understand, no semi-quantitative ultra-sonographic indices were used, this must be declared a strong limitation in the study design.
113 “The measurements were done within a second of pushing the switch. “ Please, explain why this is important.
From the initial cohort of 246 patients only less than half (117) were evaluated. How did the recruited patients compare to those of the total cohort in terms of anthropometric indices and laboratory parameters ?
Discussion: lines 169-174 may be deleted. Start directly by listing novel findings.
175 ameliorate → amelioration ? Improvement ?
Myokine may also affect → MiokineS ?
185 The above findings indicate that a change in fat to muscle mass ratio is associated with improvement of liver steatosis or stiffness. → The novel findings reported here ?
189 normal medical treatment → wher has this “normal medical treatment” been described ?
“The study limitations include not evaluating the liver steatosis or stiffness by liver biopsy” → Liver biopsy does not evaluate stiffness.
This study has many more limitations than admitted by these Authors. For example a high drop-out rate which introduces a selection bias. Ultrasonographic semi-quantitative indices which predict liver histology and correlate with metabolic parameters were not used. In addition, owing to too few individuals being recruited, separate analysis by sex was not performed, therefore neglecting recent evidence of sex differences and sex dimorphism in NAFLD.
The following conclusion: “Monitoring this index may help clinicians evaluate liver stiffness and steatosis”. is not adequately supported by data given the multiple limitations of the present study design.
Author Response
GENERAL COMMENT 1; The manuscript is quite inaccurately written (I give several specific examples below of wrong, repetitive and unclear sentences).
Response
Thank you for your recommendation. I asked native English speaker to edit our manuscript.
GENERAL COMMENT 2; The criteria for diagnosing NAFLD are wrong and these authors confuse steatosis (i.e. fat in the liver) with NAFLD (i.e. steatosis + exclusion of competing causes of steatosis).
GENERAL COMMENT 3; In addition they do not seem to use a specific laboratory protocol to rule out such competing causes of CLD.
Response
Thank you for your comments. In this study, fatty liver was diagnosed by the findings of abdominal ultrasonography performed by trained technicians. The participants with liver contrast and liver brightness among the four known criteria (hepatorenal echo contrast, liver brightness, deep attenuation and vascular blurring) were diagnosed as having fatty live. The patients who had fatty liver and who did not have exclusion criteria were defined as having NAFLD. According to your comment, we have revised the criteria for diagnosing NAFLD in the Methods section described as below.
“In this study, fatty liver was diagnosed by the findings of abdominal ultrasonography performed by trained technicians. The participants with liver contrast and liver brightness among the four known criteria (hepatorenal echo contrast, liver brightness, deep attenuation and vascular blurring) were diagnosed as having fatty live.[f] Patients were asked about the duration of diabetes, exercise habits, smoking status, the amount and types of alcoholic beverages consumed weekly during the past month. Men with a mean weekly ethanol consumption of more than 210 g, women who consumed more than 140 g/week, and patients with known liver disease were excluded.[13] Regarding known liver disease, participants who tested positive for hepatitis B antigen or hepatitis C antibody and those who reported a history of known liver disease, including viral, genetic, autoimmune, and drug-induced liver disease, were also excluded.[e] No patients had cancer or autoimmune diseases. The patients who had fatty liver and who did not have exclusion criteria were defined as having NAFLD.”
f. Hamaguchi M, Kojima T, Itoh Y, Harano Y, Fujii K, Nakajima T et al. The severity of ultrasonographic findings in nonalcoholic fatty liver disease reflects the metabolic syndrome and visceral fat accumulation. Am J Gastroenterol. Am J Gastroenterol
GENERAL COMMENT 4; The introduction is too short and the rationale is not well described.
Response
Thank you for your comment. According to your comment, we have revised the Introduction section described as below.
“Loss of body weight includes reductions of both body fat and muscle mass.[10] Sarcopenia, characterized by a decline in muscle and low muscle strength, affects clinical outcomes, including risk of cardiovascular disease and infection.[h,i] In addition., High body fat and low skeletal muscle are associated with liver steatosis and fibrosis.[4] [11][12][g] Thus, there is a possibility that not the body weight reduction, but the body fat reduction and the skeletal muscle gain is important for NAFLD. However, the effect of reduction of body fat and skeletal muscle on liver stiffness as well as steatosis has not been clarified yet.
g.Tanaka M, Okada H, Hashimoto Y, Kumagai M, Nishimura H, Oda Y, Fukui M. Relationship between nonalcoholic fatty liver disease and muscle quality as well as quantity evaluated by computed tomography. Liver Int. 2019 Sep h.Hashimoto Y, Kaji A, Sakai R, Hamaguchi M, Okada H, Ushigome E, Asano M, Yamazaki M, Fukui M. Sarcopenia is associated with blood pressure variability in older patients with type 2 diabetes: A cross-sectional study of the KAMOGAWA-DM cohort study. Geriatr Gerontol Int. 2018 Sep;18(9):1345-1349.
i.Kaji A, Hashimoto Y, Kobayashi Y, Sakai R, Okamura T, Miki A, Hamaguchi M, Kuwahata M, Yamazaki M, Fukui M. Sarcopenia is associated with tongue pressure in older patients with type 2 diabetes: A cross-sectional study of the KAMOGAWA-DM cohort study. Geriatr Gerontol Int. 2019 Feb;19(2):153-158.
GENERAL COMMENT 5; References are not invariably well taken.
Response
Thank you for your comment. According to your comment, we have revised the references.
GENERAL COMMENT 6; Some conclusions are too strong and must be softened owing to the many limitations of the study design which should be honestly admitted and discussed.
Response
Thank you for your comment. According to your comment, we have revised the Conclusions described as below.
“The reduction of fat-to-muscle-mass ratio was associated with improvement of liver stiffness but change in BMI was not. These results suggested that to improve liver steatosis and stiffness of the patients with type 2 diabetes, the physicians should focus on not only BMI, but the body compositions, such as body fat and muscle mass.”
GENERAL COMMENT 7; Editing of English is recommended.
Response
Thank you for your recommendation. I asked native English speaker to edit our manuscript.
SPECIFIC COMMENTS
SPECIFIC COMMENTS 8; Approximately 60% of diabetes patients have NAFLD--> please, specify what type of diabetes
Response
Thank you for your comment. All subjects were type 2 diabetes. I revised my manuscript and indicate it clearly.
SPECIFIC COMMENTS 9; Ref 5 in supporting the statement “many patients progress from NAFLD to liver cirrhosis and hepatocellular carcinoma” is outdated and non-specific.
Response
Thank you for your recommendation. I revised the sentence as follows; a part of patients progress from NAFLD to liver cirrhosis and hepatocellular carcinoma.[5]
Ref 5. Reid DT, Eksteen B. Murine models provide insight to the development of non-alcoholic fatty liver disease. Nutr Res Rev. 2015 Dec;28(2):133-142.
SPECIFIC COMMENTS 10; Line 35 “ NAFLD in diabetes patients to prevent further development of liver disease and to improve” → rather than “further development I would say “the progression” or “worsening” .
Response
Thank you for your kind advise. I revised the sentence as follows; NAFLD in diabetes patients to prevent the progression of liver disease and to improve.
SPECIFIC COMMENTS 11; Lines 37-40 do not match well with lines 41-44 and should probably be omitted. Conversely Lines 41-44 are well focussed on the problem but do not say and instead must say) what – based on literature studies - these authors expected to find and why. In other words the rationale should be better elucidated.
Response
Thank you for your kind advice. According to your advice, we have revised the Introduction section described as below.
“It is well known that body weight reduction leads to improvement of NAFLD.[7] In addition to that, the modification in body composition contributes to preventing life-threatening diseases.[8] Loss of body weight includes reductions of both body fat and muscle mass.[9][10] Sarcopenia, characterized by a decline in muscle and low muscle strength, affects clinical outcomes, including risk of cardiovascular disease and infection.[h,i] In addition., High body fat and low skeletal muscle are associated with liver steatosis and fibrosis.[4] [11][12][g] Thus, there is a possibility that not the body weight reduction, but the body fat reduction and the skeletal muscle gain is important for NAFLD. However, the effect of reduction of body fat and skeletal muscle on liver stiffness as well as steatosis has not been clarified yet.”
SPECIFIC COMMENTS 12; Throughout the manuscript the word non-alcoholic vs. nonalcoholic is inconsistently written. Please only one spelling (nonalcoholic).
Response
Thank you for your kind advice. I have used one spelling (nonalcoholic) instead of non-alcoholic.
SPECIFIC COMMENTS 13; Lines 52 and 54 contain the same abbreviation twice (NASH). One is enough: the first
Response
Thank you for your kind advise. I have deleted the second full spelling.
SPECIFIC COMMENTS 14; “We previously performed a two cross-sectional study in which study period was set as January 2015 57 to September 2015, or January 2015 to January 2016, respectively” sentence unclear. Is this one study or 2 studies ?
Response
Thank you for your kind comment. We previously reported the two reports from the one cohort-study, the former one used the data of January 2015 to September 2015 and the latter one used the data of January 2015 to January 2016. According to your comment, we have changed the sentences in the Methods section described as below.
“We previously reported two cross-sectional studies in which study periods were set as January 2015 to September 2015, or January 2015 to January 2016, respectively.”
SPECIFIC COMMENTS 15; Line 59 “ In this study, the research question as to clarify “ IS ?
SPECIFIC COMMENTS 16; Line 61 – Consequential → Consecutive ?
Response
Thank you for your kind your kind comment. We have revised these points.
SPECIFIC COMMENTS 17; Lines 63-64 “The patients were not involved in the recruitment to and conduct of the study.” Is this required by Ethycal Commette ? This is a quite uncommon statement.
Response
Thank you for your kind comment. It is a description mistake. We have changed the sentences as below.
“The written informed consent was obtained from all the participants.”
SPECIFIC COMMENTS 18; Line 68 The sample size was not estimated priori → A PRIORI
Response
Thank you for your kind comment. We have revised it.
SPECIFIC COMMENTS 19; The submission has many repetitions which slow down reading for example “this study is exploratory prospective study “
Response
Thank you for your kind comment. We have checked and deleted some repetitions.
SPECIFIC COMMENTS 20; Please, reword the following more concisely without any repetitions “In this study, the research question as to clarify the association of the change of body composition change with liver stiffness as well as liver steatosis was set after collecting measurements, and the study design was a retrospective analysis in a multipurpose prospective cohort study. “
SPECIFIC COMMENTS 21; “because this study is exploratory prospective study. In this retrospective cohort study, inclusion period “ In the word study is reated three times and study turns from prospective to retrospective (?)
SPECIFIC COMMENTS 22; Again, things are repeated twice here “Exclusion criteria was set as alcohol consumption, known liver disease, [7] no data 70 of lever stiffness measurement, or no data of bio impedance analysis. Men with a mean weekly ethanol consumption of more than 210 g, women who consumed more than 140 g/week, and patients with known liver disease were excluded according to the criterion of alcohol consumption”
Response
Thank you for your kind comments. We have reworded as below.
“In this study, to clarify the association of the change of body composition change with the change of liver steatosis or stiffness, the participants needed to two or more elastography and bioelectrical impedance body composition evaluations.”
“The sample size was not estimated a priori, because this is exploratory study.”
“We included the patients who performed both elastography and bioelectrical impedance body composition evaluations. We excluded the patients without NAFLD, the definition was described below”
SPECIFIC COMMENTS 23; English is poor “study population was consisted with” “Researchers who are specialized in diabetology are conducted in clinical practice in this cohort “
Response
Thank you for your kind comments. We have reworded as below.
“the patients, with two or more elastography and bioelectrical impedance body composition evaluations, were extracted.”
SPECIFIC COMMENTS 24; Lines 86-87 “Men with a mean weekly ethanol consumption of more than 210 g, women who consumed more than 140 g/week, and patients with known liver disease were excluded.” exactly duplicates lines 71 and 72.
Response
Thank you for your kind comments. We have deleted the line 71 and 72.
SPECIFIC COMMENTS 25; Throughout the manuscript “Known liver disease”--> What does this mean ? Based on medical history? No laboratory test to rule out competing etiologies. No questions on potentially steatogenic drug treatment ?
SPECIFIC COMMENTS 26; “NAFLD was diagnosed by abdominal ultrasonography” This is conceptually wrong. Ultrasonography cannot see NAFLD although it can appreciate the presence of STEATOSIS. IF, as I understand, no semi-quantitative ultra-sonographic indices were used, this must be declared a strong limitation in the study design.
Response
Thank you for your comments. According to your General comment 2, we have revised the Methods section described as below.
“In this study, fatty liver was diagnosed by the findings of abdominal ultrasonography performed by trained technicians. The participants with liver contrast and liver brightness among the four known criteria (hepatorenal echo contrast, liver brightness, deep attenuation and vascular blurring) were diagnosed as having fatty live.[f] Patients were asked about the duration of diabetes, exercise habits, smoking status, the amount and types of alcoholic beverages consumed weekly during the past month. Men with a mean weekly ethanol consumption of more than 210 g, women who consumed more than 140 g/week, and patients with known liver disease were excluded.[13] Regarding known liver disease, participants who tested positive for hepatitis B antigen or hepatitis C antibody and those who reported a history of known liver disease, including viral, genetic, autoimmune, and drug-induced liver disease, were also excluded.[e] No patients had cancer or autoimmune diseases. The patients who had fatty liver and who did not have exclusion criteria were defined as having NAFLD.”
SPECIFIC COMMENTS 26; 113 “The measurements were done within a second of pushing the switch. “ Please, explain why this is important.
Response
Thank you for your comment. We have deleted this sentence, because of no important meaning.
SPECIFIC COMMENTS 27; From the initial cohort of 246 patients only less than half (117) were evaluated. How did the recruited patients compare to those of the total cohort in terms of anthropometric indices and laboratory parameters ?
Response
Thank you for your comment. As you say, to compare the data of the patients who did not perform the follow up and that of the patients who did is important. According to your comment, we have analyzed the difference of anthropometric indices and laboratory parameters in the patients with and without performing follow-up. There was no difference between the patients with and without performing follow-up. We have added this result in the Results section and supplemental Table.
Also we added the following sentence;
“We have analyzed the difference of anthropometric indices and laboratory parameters in the patients with and without performing follow-up (Supplemental Table). There was no difference between the patients with and without performing follow-up.”
SPECIFIC COMMENTS 28; Discussion: lines 169-174 may be deleted. Start directly by listing novel findings.
Response
Thank you for your comment. According to your comment, we have revised the Discussion section described as below.
“This study investigated the association of metabolic or physical variables and improvement of liver stiffness as well as steatosis in type 2 diabetes patients with NAFLD. We revealed that the reduction of fat to muscle mass ratio was associated with improvement of liver stiffness, but change in BMI was not associated with improvement of liver stiffness.”
SPECIFIC COMMENTS 29; 175 ameliorate → amelioration ? Improvement ?
SPECIFIC COMMENTS 30; Myokine may also affect → MiokineS ?
Response
Thank you for your comments. We have changed these words.
SPECIFIC COMMENTS 31; 185 The above findings indicate that a change in fat to muscle mass ratio is associated with improvement of liver steatosis or stiffness. → The novel findings reported here ?
Response
Thank you for your comments. According to your comment, we have revised the sentence “The above findings indicate that a change in fat to muscle mass ratio is associated with improvement of liver steatosis or stiffness” into “The above findings support our results”.
SPECIFIC COMMENTS 32; 189 normal medical treatment → where has this “normal medical treatment” been described ?
Response
Thank you for your comments. We have changed the sentence as below.
We added the sentence “neither diet therapy nor exercise therapy have been changed during this one-year period.” Instead of the sentence of “normal medical treatment …”
SPECIFIC COMMENTS 32; “The study limitations include not evaluating the liver steatosis or stiffness by liver biopsy” → Liver biopsy does not evaluate stiffness.
Response
Thank you for your comment. As you say, liver biopsy dose not evaluate stiffness, but fibrosis. We have revised this point as below.
“The study limitations include not evaluating the liver steatosis or fibrosis by liver biopsy, even though transient elastography has been validated for diagnosis of liver steatosis[33] or stiffness.[34]”
SPECIFIC COMMENTS 33; This study has many more limitations than admitted by these Authors. For example a high drop-out rate which introduces a selection bias. Ultrasonographic semi-quantitative indices which predict liver histology and correlate with metabolic parameters were not used. In addition, owing to too few individuals being recruited, separate analysis by sex was not performed, therefore neglecting recent evidence of sex differences and sex dimorphism in NAFLD.
Response
Thank you for your comments. As you say, these points were limitations of our study. According to your comments, we have added these points in the Limitations.
SPECIFIC COMMENTS 34; The following conclusion: “Monitoring this index may help clinicians evaluate liver stiffness and steatosis”. is not adequately supported by data given the multiple limitations of the present study design.
Response
Thank you for your comment. According to your comment, we have changed the sentence as below.
“These results suggested that to improve liver steatosis and stiffness of the patients with type 2 diabetes, the physicians should focus on not only BMI, but the body compositions, such as body fat and muscle mass.”
Round 2
Reviewer 2 Report
Thank you for the revised version. The article is much easier to read after the English editing, however there are still some minor mistakes, as follows: (please correct):
line 58: the participants needed to two or more elastography and bioelectrical - there should be no "to"
line 65: patients who performed both elastography and bioelectrical impedance body composition - patients did not perform the examination, they underwent examinatios
line 115: the patients who change from normal to extensive liver stiffness - changed, not change
lines 164, 166, 180, 182 - changed, not change
line 176 - "live stiffness status" - liver?
lines 177, 178 - The change in fat to muscle ratio of group 3 was higher than that of group 1 (p= 0.038, by Tukey-Kramert HSD test) and group 1 (p = 0.048, by Tukey-Kramert HSD test). - basing on the figure, I think the difference was between group 0 and 3 and 1 and 3, please verify
line 244 and 245: "A high drop-out rate which introduces a selection bias." - I think there is something missing in this sentence.
Author Response
Thank you very much for sparing your precious time in reviewing our paper, and for your email dated 26 Dec 2019 enclosing reviewers’ comments. We thank reviewers for their thoughtful remarks and advices, which have greatly helped us to revise and improve our paper.
We hope the revised version is now suitable for publication and look forward to hearing from you in due course.
Yours Sincerely,
Michiaki Fukui, MD, PhD
Department of Endocrinology and Metabolism, Kyoto Prefectural University of Medicine, Graduate School of Medical Science
Address: 465 Kajii-cho, Kawaramachi-Hirokoji, Kamigyo-ku, Kyoto 602-8566, Japan
Fax: +81-75-252-3721, Tel: +81-75-251-5505
E-mail: michiaki@koto.kpu-m.ac.jp
Response to Reviewer 2
Comment 1
line 58: the participants needed to two or more elastography and bioelectrical - there should be no "to"
Response 1
Thank you for your comment. I deleted "to" from the sentence.
Comment 2
line 65: patients who performed both elastography and bioelectrical impedance body composition - patients did not perform the examination, they underwent examinatios
Response 2
Thank you for your comment. I revised the sentence as follows; We included the patients who underwent both elastography and bioelectrical impedance body composition evaluations.
Comment 3
line 115: the patients who change from normal to extensive liver stiffness - changed, not change
Response 3
Thank you for your comment. I revised the sentence as follows; group 2, the patients who changed from normal to extensive liver stiffness.
Comment 4
lines 164, 166, 180, 182 - changed, not change
Response 4
Thank you for your comment. I revised “change” into “changed”.
Comment 5
line 176 - "live stiffness status" - liver?
Response 5
Thank you for your comment. I revised “live” into “liver”.
Comment 6
lines 177, 178 - The change in fat to muscle ratio of group 3 was higher than that of group 1 (p= 0.038, by Tukey-Kramert HSD test) and group 1 (p = 0.048, by Tukey-Kramert HSD test). - basing on the figure, I think the difference was between group 0 and 3 and 1 and 3, please verify
Response 6
Thank you for your comment. According to your comment, we have reanalyzed the data. The change in fat to muscle ratio of group 4 was higher than that of group 3 (p <0.001, by Tukey-Kramert HSD test), that of group 2 (p <0.001, by Tukey-Kramert HSD test) and group 1 (p <0.001, by Tukey-Kramert HSD test). The change in fat to muscle ratio of group 1 was higher than that of group 2 (p = 0.036, by Tukey-Kramert HSD test). We have revised the Result section and the Figure 2 described as below.
“The change in fat to muscle ratio was -4.03 (30.7) % in group 1, -40.6 (12.4) % in group 2, -14.4 (34.6) % in group 3 and 117.2 (180.3) % in group 4 (p <0.001) and that of group 3 was higher than that of group 3 (p <0.001), that of group 2 (p <0.001) and group 1 (p <0.001).”
Comment 7
line 244 and 245: "A high drop-out rate which introduces a selection bias." - I think there is something missing in this sentence.
Response 6
Thank you for your comment. According to your comment, we have revised the limitation described as below.
“A high dropout rate might be the cause of a selection bias.”